# Economic and modeling evidence for tuberculosis preventive therapy among people living with HIV: A systematic review and meta-analysis

**Aashna Uppal**[1,2,3], **Samiha Rahman**[1,2,3], **Jonathon R. Campbell**[1,2,3], **Olivia Oxlade**[3], **Dick Menzies**[1,2,3]*

1 Montréal Chest Institute, Montréal, Québec, Canada, 2 Respiratory Epidemiology and Clinical Research Unit, Centre for Outcomes Research and Evaluation, Research Institute of McGill University Health Centre, Montréal, Québec, Canada, 3 McGill International Tuberculosis Centre, Montréal, Québec, Canada

* dick.menzies@mcgill.ca

## Abstract

**Data Availability Statement:** All relevant data are within the manuscript and its Supporting Information files.

### Background

Human immunodeficiency virus (HIV) is the strongest known risk factor for tuberculosis (TB) through its impairment of T-cell immunity. Tuberculosis preventive treatment (TPT) is recommended for people living with HIV (PLHIV) by the World Health Organization, as it significantly reduces the risk of developing TB disease. We conducted a systematic review and meta-analysis of modeling studies to summarize projected costs, risks, benefits, and impacts of TPT use among PLHIV on TB-related outcomes.

### Methods and findings

We searched MEDLINE, Embase, and Web of Science from inception until December 31, 2020. Two reviewers independently screened titles, abstracts, and full texts; extracted data; and assessed quality. Extracted data were summarized using descriptive analysis. We performed quantile regression and random effects meta-analysis to describe trends in cost, effectiveness, and cost-effectiveness outcomes across studies and identified key determinants of these outcomes. Our search identified 6,615 titles; 61 full texts were included in the final review. Of the 61 included studies, 31 reported both cost and effectiveness outcomes. A total of 41 were set in low- and middle-income countries (LMICs), while 12 were set in high-income countries (HICs); 2 were set in both. Most studies considered isoniazid (INH)-based regimens 6 to 2 months long (*n* = 45), or longer than 12 months (*n* = 11). Model parameters and assumptions varied widely between studies. Despite this, all studies found that providing TPT to PLHIV was predicted to be effective at averting TB disease. No TPT regimen was substantially more effective at averting TB disease than any other. The cost of providing TPT and subsequent downstream costs (e.g. post-TPT health systems costs) were estimated to be less than $1,500 (2020 USD) per person in 85% of studies that reported cost outcomes (*n* = 36), regardless of study setting. All cost-effectiveness analyses

**Funding:** This work was funded by the Bill & Melinda Gates Foundation (Grant Number INV-003634). The initial study questions for the papers included in the PLOS Collection were drafted together with input from staff of the Bill & Melinda Gates Foundation, but they had no further role in study design, data collection and analysis, decision to publish, or preparation of the manuscript.

**Competing interests:** The authors have declared that no competing interests exist.

**Abbreviations:** ART, antiretroviral therapy; CI, confidence interval; DALY, disability-adjusted life year; GDP, gross domestic product; HIC, high-income country; HIV, human immunodeficiency virus; ICER, incremental cost-effectiveness ratio; IGRA, interferon gamma release assay; INH, isoniazid; INMB, INmune Bio; IQR, interquartile range; LMIC, low- and middle-income country; LTBI, latent tuberculosis infection; PLHIV, people living with HIV; PRISMA, Preferred Reporting Items for Systematic Reviews and Meta-Analyses; PSA, probabilistic sensitivity analysis; QALY, quality-adjusted life year; TB, tuberculosis; TPT, tuberculosis preventive treatment; TST, tuberculin skin test.

concluded that providing TPT to PLHIV was potentially cost-effective compared to not providing TPT. In quantitative analyses, country income classification, consideration of antiretroviral therapy (ART) use, and TPT regimen use significantly impacted cost-effectiveness. Studies evaluating TPT in HICs suggested that TPT may be more effective at preventing TB disease than studies evaluating TPT in LMICs; pooled incremental net monetary benefit, given a willingness-to-pay threshold of country-level per capita gross domestic product (GDP), was $271 in LMICs (95% confidence interval [CI] −$81 to $622, $p = 0.12$) and was $2,568 in HICs (−$32,115 to $37,251, $p = 0.52$). Similarly, TPT appeared to be more effective at averting TB disease in HICs; pooled percent reduction in active TB incidence was 20% (13% to 27%, $p < 0.001$) in LMICs and 37% (−34% to 100%, $p = 0.13$) in HICs. Key limitations of this review included the heterogeneity of input parameters and assumptions from included studies, which limited pooling of effect estimates, inconsistent reporting of model parameters, which limited sample sizes of quantitative analyses, and database bias toward English publications.

## Conclusions

The body of literature related to modeling TPT among PLHIV is large and heterogeneous, making comparisons across studies difficult. Despite this variability, all studies in all settings concluded that providing TPT to PLHIV is potentially effective and cost-effective for preventing TB disease.

## Author summary

### Why was this study done?

- Human immunodeficiency virus (HIV) is the strongest know risk factor for tuberculosis (TB). While the uptake of tuberculosis preventive treatment (TPT) has increased in recent years, a summarization of costs, risks, benefits, and impacts of TPT among people living with HIV (PLHIV) does not exist.

### What did the researchers do and find?

- We conducted a systematic review and meta-analysis to synthesize data on costs and cost-effectiveness, as well as risks and impact on TB morbidity and mortality associated with TPT provided to PLHIV. Our search identified 6,615 titles; 61 full texts were included in the final review. We performed quantile regression and a random effects meta-analysis to describe key trends in cost, effectiveness, and cost-effectiveness outcomes across studies and to identify key determinants of these outcomes.

- Values of model parameters varied widely. In our quantile regression analyses, we found that TPT appeared to be more effective reducing at active TB incidence and more cost-effective in high-income countries (HICs), compared to low- and middle-income countries (LMICs).

- The pooled incremental net monetary benefit, given a willingness-to-pay threshold of country-level gross domestic product (GDP) per capita, was positive for both LMICs and HICs, meaning that TPT was potentially cost-effective compared to no TPT, regardless of study setting.

- Aside from our quantitative results, individual study conclusions found that providing TPT to PLHIV was predicted to be effective at averting TB disease and was predicted to be cost-effective compared to not providing TPT.

### What do these findings mean?

- Heterogeneity and inconsistent reporting of model parameters made it difficult to summarize these studies and limited the extent of pooling through meta-analytical techniques. This underscores the need for better standardization of models for TB.

- The findings of this review support greater resource allocation in all settings to expand programs that deliver TPT to PLHIV.

## Background

Until 2020, tuberculosis (TB) was the leading cause of death due to an infectious disease, causing an estimated 1.4 million deaths in 2019 [1]. Human immunodeficiency virus (HIV) is the strongest known risk factor for TB through its impairment of T-cell immunity [2]. People living with HIV (PLHIV), without the use of antiretroviral therapy (ART), are 20 to 30 times more likely to progress to TB disease than those without HIV [2]. Approaches to reduce TB morbidity and mortality in PLHIV include provision of effective ART and providing tuberculosis preventive treatment (TPT) to those who may be infected with and are at risk of *Mycobacterium tuberculosis* [3].

TPT is strongly recommended for PLHIV by the World Health Organization because it is known to significantly reduce the risk of developing TB disease. [4]. Uptake of TPT has increased substantially in recent years, with close to 2-fold (1.8 million to 3.6 million between 2018 and 2019) increase in TPT initiation among PLHIV [1]. To further advance the uptake of TPT, understanding the potential benefits and the full costs of TPT among PLHIV is necessary to inform decision-makers and plan effective service delivery. From a health system perspective, drug regimen costs are easy to quantify, but the total costs associated with provision of preventive care (e.g., lab testing and personnel costs), as well as the epidemiologic impact, are more difficult to quantify. Transmission modeling and cost-effectiveness studies can provide part of the key evidence needed to efficiently improve TPT coverage and service delivery [5].

The objective of this study was to systematically review published literature to synthesize the costs and cost-effectiveness, as well as risks and impacts on TB morbidity and mortality associated with TPT provided to PLHIV.

## Methods

This review was done in accordance with the Preferred Reporting Items for Systematic Reviews and Meta-Analyses (PRISMA) guidelines; see S1 PRISMA Checklist for details [6].

We prospectively registered this review in PROSPERO under the registration ID CRD42020187934.

## Search strategy

We searched MEDLINE, Embase, and Web of Science from database inception to December 31, 2020. The search strategy is described in detail in Table A in S1 Text. References of included studies were also searched for other relevant literature. Two reviewers independently screened titles and abstracts and then full texts to identify additional studies. Studies were included if they met the following criteria: (1) study population included at least a subset of PLHIV in whom active TB had been excluded; (2) study considered at least 1 TPT regimen (6 isoniazid [INH], 9 INH, etc.); (3) study investigated costs, cost-effectiveness (assessed for programmatic yields such as case detection or extended to utility outcomes such as disability-adjusted life years [DALYs] or quality-adjusted life years [QALYs]), and/or epidemiologic impact estimates such as TB incidence; (4) if PLHIV were a subset of the study population, outcomes were disaggregated by HIV status; (5) study outcomes were disaggregated by receipt of TPT; (6) study design was either economic evaluation (namely cost analysis, econometric analysis, or cost-effectiveness analysis), or a mathematical or epidemiologic modeling study; and (7) full text was available.

## Data extraction and quality assessment

Data extracted from each study included study design, setting, data sources, cohort type (PLHIV or general population with PLHIV subset), TPT regimen given, values of selected key model parameters, and projected outcomes over the analytic period among those who did and did not receive TPT (see Table R in S1 Text for extraction form). For cost-effectiveness studies that compared PLHIV receiving TPT to those not receiving TPT, incremental cost-effectiveness ratios (ICERs) were also extracted. Lastly, results from any sensitivity or threshold analyses that were performed were extracted.

Included studies underwent quality assessment using a 10-item quality assessment checklist based on quality assessment guides from the Panel on Cost-Effectiveness in Health and Medicine for Economic Evaluations and the International Society for Pharmacoeconomics and Outcomes Research & Society for Medical Decision Making's Modeling Good Research Practices Task Force for Modelling Studies [7,8] (see Table B in S1 Text for list of criteria). Two items were considered essential to ratings of high quality: (1) the study included a clear description of the interventions under evaluation; and (2) the study used an appropriate source to inform at least one of the following key input parameters (if included in the model): rate of TPT completion/adherence, TPT efficacy in preventing active TB, and tuberculin skin test (TST)/interferon gamma release assay (IGRA) test sensitivity and specificity. An appropriate source was a systematic review or meta-analysis, with some exceptions; see Table B in S1 Text for details. A high-quality study met both items and met at least 4 of the other 8 items included in the quality assessment. Studies that did not meet these criteria were considered low quality.

## Data analysis

Descriptive analyses qualitatively summarized study data, stratified by country-level income for the population considered in the study (low- and middle-income countries [LMICs] and high-income countries [HICs], as defined by the World Bank). Key input parameters collated were country-level income, consideration of ART use (i.e., whether or not the study included a parameter related to ART efficacy or ART costs), TPT regimen given, latent tuberculosis infection (LTBI) prevalence, modeled length of follow-up (i.e., time horizon), TPT efficacy in

preventing active disease, level of adherence to TPT, and the probability of a fatal adverse event due to TPT. We summarized 2 key outcomes: (1) relative reduction in active TB incidence between PLHIV given and not given TPT; and (2) incremental net monetary benefit comparing PLHIV given and not given TPT, which was standardized to 2020 USD [9]. We calculated incremental net monetary benefit as follows:

$$Difference\ in\ Effectiveness * Willingness\ to\ Pay - Difference\ in\ Cost$$

Conventionally, utility values (such as DALYs) are used to complete the net monetary benefit calculation; however, we considered relative reduction in active TB incidence as a measure of effectiveness instead because only 6 studies reported DALYs. We assumed a willingness-to-pay threshold of country-level gross domestic product (GDP) per capita and explored this assumption's effects with sensitivity analyses. Net monetary benefit was chosen over ICERs because of ease of interpretation (i.e., the magnitude of negative ICERs does not convey useful information) [10] and proportionality to scale vis-à-vis the willingness-to-pay threshold.

Subsequent quantile regression analysis to estimate the median of the target value (instead of the mean) was used to explore the extent to which key input parameters were associated with key outcomes. Both univariable and multivariable quantile regression analyses were performed. Quantile regression was chosen over simple linear regression because our data were unlikely to meet assumptions required for linear regression. We included all categorical variables (country-level income, ART use, and TPT regimen given) in multivariable analyses and subsequently added combinations of continuous variables, as long as the sample size did not drop below 10 and there was no strong correlation (i.e., the correlation coefficient was not close to or equal to 1 or −1) between independent variables included in the model. To ensure a consistent sample size among final multivariable models, missing values for independent variables were imputed using medians.

We also performed meta-analysis on our 2 key outcomes. As there is no universal method to meta-analyze results of modeling studies, we referred to other publications' methodology, which included estimating standard error from probabilistic sensitivity analyses (PSAs) or one-way analyses if PSA was not done [11]. We pooled studies conservatively using an inverse variance method, with a Sidik–Jonkman estimator for tau and Hartung–Knapp adjustment for our random effects model. We pooled studies conducted in HICs and LMICs separately and where possible, further stratified on key input parameters.

## Results

Our initial search identified 6,615 titles. One additional article was identified from the reference list of an included article. After titles and abstracts were reviewed, 104 were selected for full text review, of which 61 [12–72] met the study inclusion criteria (Fig 1).

### Study characteristics and quality assessment

Study characteristics varied widely, and input parameters included in modeling studies were clinically heterogeneous. Study characteristics are summarized in Table 1, with additional information in Table F in S1 Text. Fifty-four studies (out of 61) used modeling methods to evaluate impact and/or cost-effectiveness of TPT in PLHIV; 28 used modeling methods that excluded TB transmission; and 26 used modeling methods that included transmission. Seven studies that did not use modeling methods were either cost analyses conducted alongside clinical trials (*n* = 5) or cost-effectiveness evaluations conducted alongside observational studies (*n* = 2). Thirty-six studies (59%) reported both cost and effectiveness or utility outcomes, 25 (41%) reported effectiveness or utility outcomes only, and 5 (8%) reported cost outcomes only. Forty-one (67%) studies were set

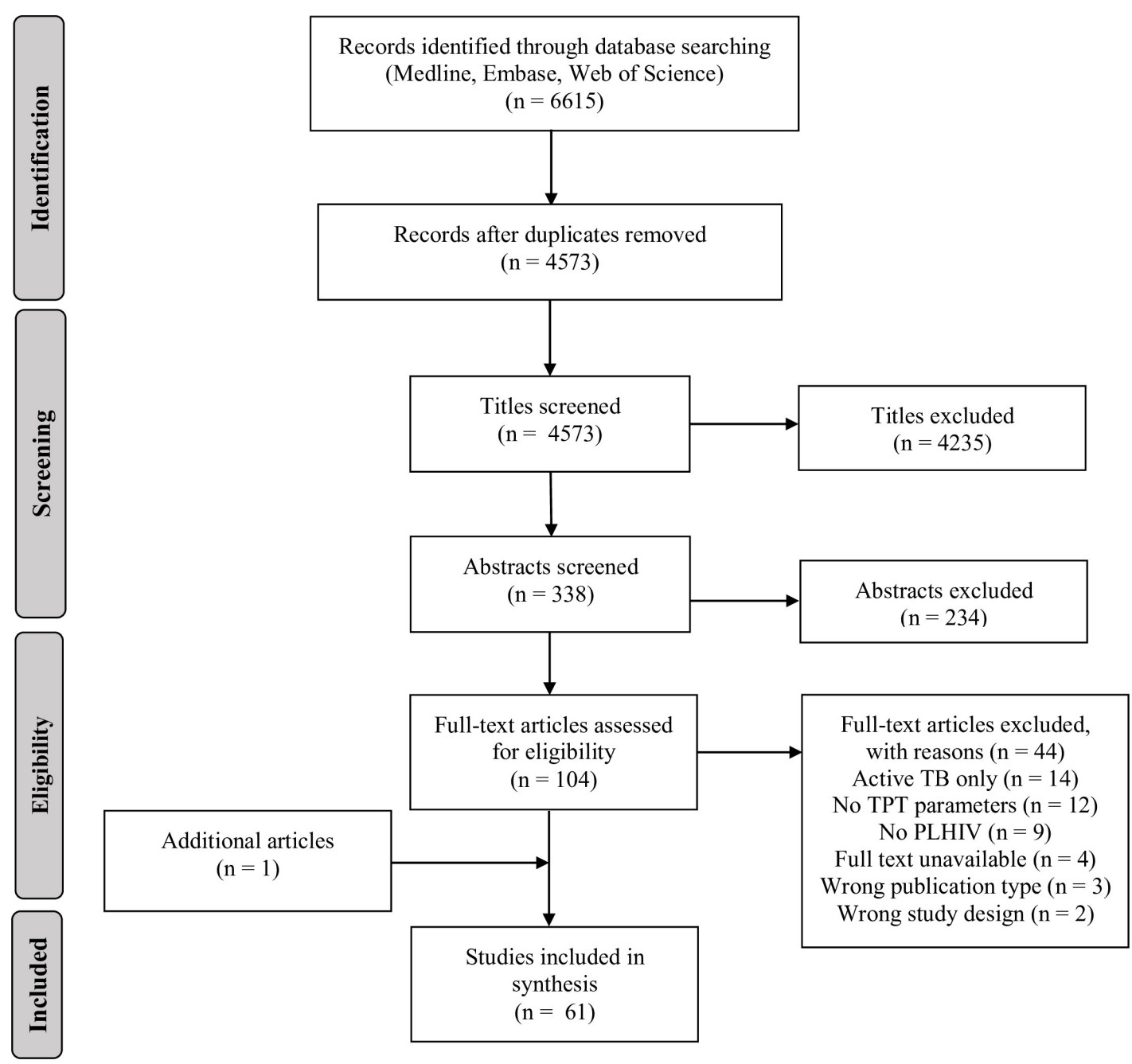

*From:* Moher D, Liberati A, Tetzlaff J, Altman DG, The PRISMA Group (2009). *P*referred *R*eporting *I*tems for *S*ystematic Reviews and *M*eta-*A*nalyses: The PRISMA Statement. PLoS Med 6(7): e1000097. doi:10.1371/journal.pmed1000097

**For more information, visit www.prisma-statement.org.**

**Fig 1. Selection of included studies.** PLHIV, people living with HIV; TB, tuberculosis; TPT, tuberculosis preventive therapy. Citation: Moher D, Liberati A, Tetzlaff J, Altman DG, The PRISMA Group. Preferred Reporting Items for Systematic Reviews and Meta-Analyses: The PRISMA Statement. PLoS Med. 2009;6(7):e1000097. doi: 10.1371/journal.pmed1000097.

**Table 1. Study characteristics.**

| Factor/parameter | Number of studies |
|---|---|
| Total number of studies included | 61 |
| **Model characteristics** | |
| **Type of outcome reported** | |
| Cost and effectiveness/utility outcomes | 31 |
| Effectiveness/utility outcomes only | 25 |
| Cost outcomes only | 5 |
| *Effectiveness outcomes reported (3 most common)* | |
| Active TB cases/incidence/prevalence | 46 |
| TB-related or TPT-related deaths/mortality | 24 |
| TPT-related hepatotoxicity* | 11 |
| *Utility outcomes reported* | |
| QALYs | 8 |
| DALYs | 6 |
| **Modeling method (only modeling studies** ** , *N* = 54) | |
| Modeling analysis excludes disease transmission | 28 |
| Modeling analysis includes disease transmission | 26 |
| **Costing details‡ (only studies that report cost outcomes, *N* = 38)** | |
| *Costing method* | |
| Mixed methods, i.e., study obtained cost parameters from both primary and secondary data sources | 19 |
| Empiric costing, i.e., study obtained cost parameters exclusively from primary data sources | 10 |
| Simple costing, i.e., study obtained cost parameters exclusively from secondary data sources | 9 |
| *Costing perspective* | |
| Health system or health provider | 36 |
| Societal | 1 |
| Patient | 1 |
| **Analytic horizon** | |
| No analytic horizon stated or analytic horizon <2 years | 5 |
| Analytic horizon 2 to 10 years | 33 |
| Analytic horizon >10 years‡‡ | 23 |
| **Population** | |
| Main population investigated was PLHIV | 33 |
| Pregnant women living with HIV | 3 |
| Main population investigated was general population with PLHIV subset | 22 |
| Main population investigated was people who use drugs with PLHIV subset | 3 |
| Main population investigated was people experiencing homelessness with PLHIV subset | 1 |
| Main population investigated was gold mine workers with PLHIV subset | 1 |
| **Study setting** | |
| *World Bank income classification* | |
| LMIC | 41 |
| HIC | 12 |
| Multiple settings, both LMIC and HIC | 2 |
| *World Health Organization regions* | |
| African Region | 33 |
| Region of the Americas | 18 |
| Southeast Asian Region | 6 |
| Western Pacific Region | 2 |

(*Continued*)

**Table 1.** (Continued)

| Factor/parameter | Number of studies |
|---|---|
| European Region | 1 |
| Eastern Mediterranean Region | 0 |
| *No specific setting* | 6 |
| **Intervention characteristics** | |
| **Used or did not use LTBI tests to determine if TPT indicated** | |
| Did not use LTBI tests to determine if TPT indicated | 25 |
| Used LTBI tests to determine if TPT indicated[†] | 19 |
| Compared using LTBI tests and not using LTBI tests to determine if TPT indicated | 16 |
| *LTBI test modeled* | |
| TST | 29 |
| IGRA | 7 |
| Compared the impact/effectiveness/cost-effectiveness of TST versus IGRA | 3 |
| **TPT regimens modeled[††]** | |
| *INH monotherapy* | |
| 6 to 12 months INH | 45 |
| >12 months INH | 11 |
| INH, duration not specified | 12 |
| *Rifamycin-containing regimens* | |
| 3 months INH and RPT | 6 |
| 3 months INH and RIF | 2 |
| 1 month INH and RPT | 1 |
| *TPT, regimen not specified* | 2 |
| **HIV treatment** | |
| ART use (cost and/or impact) considered in the model's base case | 33 |
| Considered ART coverage/impact in sensitivity analysis (in addition to base case) | 13 |
| Disaggregated TB- or TPT-related input parameters by ART status | 12 |
| Disaggregated TPT-related outcomes by ART status | 5 |

[*] This includes all studies that reported TPT-related hepatotoxicity as well as 2 studies that reported the following: (1) liver function abnormality; and (2) drug induced liver injury.

[**] Seven studies were not modeling studies but had desired outcomes, for example, costing analyses done alongside observational studies.

[‡] Primary data sources are clinical trials, regional programs or clinics, government reports or data, interviews with clinic staff, program evaluations, and hospital or clinic records. Studies that utilized data from local trials, regional programs or clinics, or local program evaluations were those that included the cost of program implementation. Secondary data sources are published literature or unpublished reports (e.g., NGO reports). Studies tended to utilize a mix of primary and secondary data sources for cost parameters.

[‡‡] The longest time horizon was 100 years.

[†] "Indicated" is when TPT is given to a subset of the cohort, based on a positive LTBI test (IGRA or TST).

[††] The numbers beside each regimen indicate the number of months that regimen was given.

ART, antiretroviral therapy; DALY, disability-adjusted life year; HIC, high-income country; HIV, human immunodeficiency virus; IGRA, interferon gamma release assay; INH, isoniazid; LMIC, low- and middle-income country; LTBI, latent tuberculosis infection; NGO, nongovernmental organization; PLHIV, people living with HIV; QALY, quality-adjusted life year; RIF, rifampin; RPT, rifapentine; TB, tuberculosis; TPT, tuberculosis preventative therapy; TST, tuberculin skin test.

in LMICs, of which 33 (80%) studies were set in the African Region. Forty-five (74%) studies evaluated 6 to 12 months of daily INH; only 9 (15%) studies considered rifamycin-based regimens for TPT. Thirty-six (59%) studies explored the use of TST or IGRA to guide the decision to recommend TPT (i.e., those with a positive TST or IGRA were provided TPT).

Model parameters and data sources for these parameters are summarized in Table D in S1 Text. Although values used for input parameters varied widely, as seen in Table E in S1 Text, there did not appear to be any important differences between parameters that were based on published data, compared to parameters based on assumptions (i.e., no sources or references cited for the values used).

Of the 61 studies included in this review, 51 were classified as high quality and 10 as low quality. The detailed quality assessments are shown in Tables B and C in S1 Text.

## Projected outcomes and study conclusions

In all studies that reported effectiveness or utility outcomes, compared to no TPT, the provision of TPT for PLHIV was more effective at reducing active TB incidence, TB-related mortality, and DALYs and was more effective at increasing QALYs and life expectancy (Tables G and H in S1 Text).

There were 68 unique strategies within studies that reported a relative reduction in active TB incidence comparing PLHIV given TPT to PLHIV not given TPT, which also specified a TPT regimen. Modeling strategies among these studies were heterogeneous, as were model parameters. Twenty-six studies considered TB transmission, and 42 did not consider TB transmission. The median TPT efficacy in preventing active TB ranged from 0.11 to 1 and percent completion of TPT ranged from 7% to 100%, for example. As seen in Fig 2, the relative reduction in active TB incidence with TPT compared to no TPT ranged from nearly 0% to nearly 100%. There was no apparent effect on reduction in active TB incidence between modeling studies that considered versus did not consider TB transmission; the median percent reduction in active TB incidence was 28% (interquartile range [IQR] 19% to 51%) among studies modeling without TB transmission and 28% (IQR 11% to 70%) among studies modeling with TB

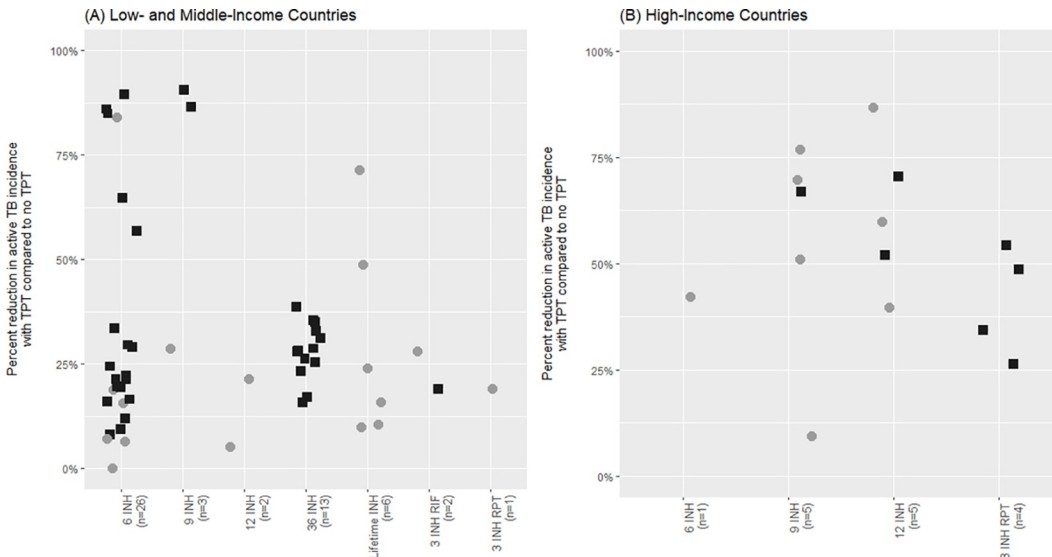

**Fig 2. Percent reduction in active TB incidence in studies comparing TPT versus no TPT, by TPT regimen and country-level income.** INH, isoniazid; RIF, rifampin; RPT, rifapentine; TB, tuberculosis; TPT, tuberculosis preventive therapy. Each data point represents a study arm. Modeling methods are distinguished in this figure; filled black squares (■) represent decision analysis models, while filled gray circles (●) represent transmission models.

transmission. On the other hand, the median percent reduction in active TB incidence was 28% (IQR 17% to 50%) in LMICs, whereas it was 48% (IQR 20% to 63%) in HICs. The latter association is further explored in regression analyses.

Of the 38 studies that reported cost outcomes, 9 obtained cost parameters exclusively from secondary data sources, while 10 employed empiric costing methods, gathering cost parameters exclusively from primary data sources (Table 1). Of those that undertook empiric costing or a combination of simple and empiric costing, 15 studies included costs associated with TPT implementation. Most of these 38 studies were modeling studies that excluded transmission (*n* = 30), while a minority were transmission modeling studies (*n* = 4) or analyses conducted alongside clinical trials or observational studies (*n* = 4). Within these 38 studies, parameters differed widely; TPT efficacy in averting active TB ranged from 0.11 to 1, time horizon ranged from 1 to 100 years, and LTBI prevalence ranged from 0.03 to 1. The per-person costs of TPT did not vary greatly by regimen, regardless of country-level income, as seen in Fig 3. The strategies included in Fig 3 (*n* = 63) compared TPT to no TPT and included the downstream health systems costs related to active TB care. The median per-person cost was $299 (IQR $73 to $756) among these strategies. Six other strategies also reported per-person costs for TPT, but excluded downstream health systems costs related to active TB care. The median per-person cost was $148 (IQR $117 to $579) among these 6 strategies. The use of ART also contributed to the magnitude of per-person costs of TPT; the median cost among studies that considered the cost of ART was $592 (IQR $152 to $756), whereas the median cost among studies that did not consider the cost of ART was $195 (IQR $65 to $365).

Of the 47 unique strategies within studies that reported incremental cost per active TB case averted, 35 were set in LMICs, while 12 were set in HICs. Values of model parameters varied widely; the median LTBI prevalence was 0.26 (range 0.02 to 0.64), the median time horizon was 3 years (range 1 to 20 years), and the median TPT efficacy was 0.49 (range 0.11 to 0.90).

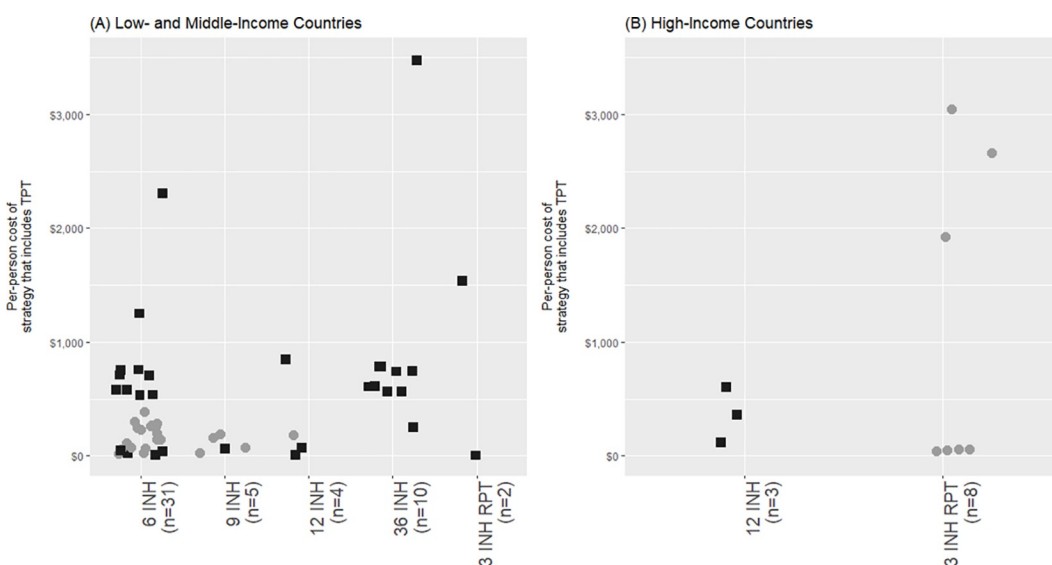

**Fig 3. Per-person cost of TPT versus no TPT, by TPT regimen and country-level income.** ART, antiretroviral therapy; INH, isoniazid; RIF, rifampin; RPT, rifapentine; TB, tuberculosis; TPT, tuberculosis preventive therapy. Each data point represents a study arm or "strategy." Outliers are analyzed in further detail in the Outliers section of S1 Text. Costs displayed in this figure include program costs related to TPT delivery (drug costs, personnel costs, and material costs) as well as costs related to TB care for those who develop active TB (drug costs, hospitalization costs, and personnel costs). Importantly, these come from studies that compared the use of TPT to no TPT and do not include studies that comparing directed TPT to TPT for all. The use of ART is distinguished in this figure; filled black squares (■) represent strategies that included the cost of ART, while filled gray circles (●) represent strategies that did not include the cost of ART.

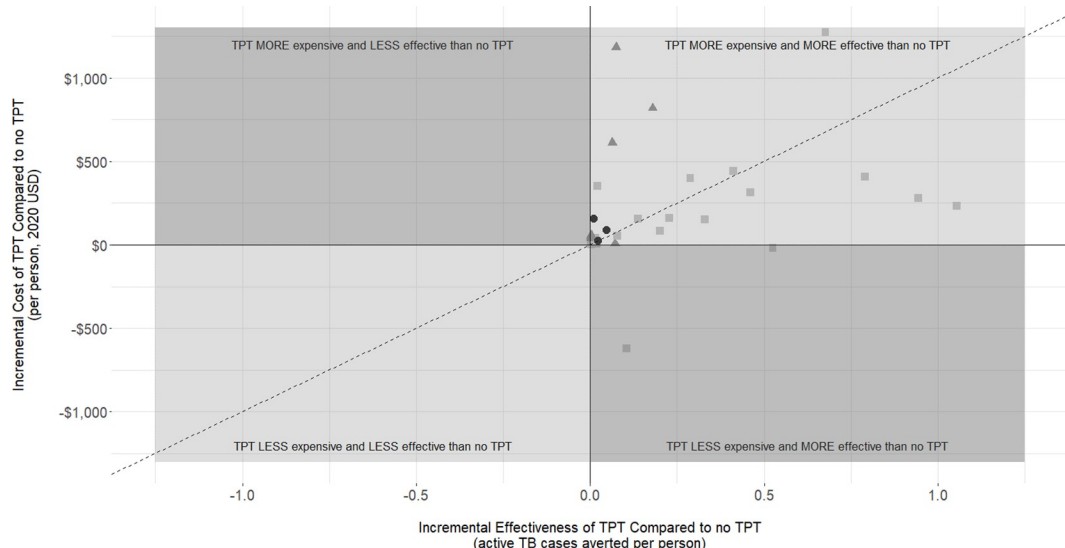

**Fig 4. Incremental cost versus incremental effectiveness, comparing TPT to no TPT.** INH, isoniazid; TB, tuberculosis; TPT, tuberculosis preventive therapy; USD, United States dollar. Each data point represents an individual study arm. Data points in the top right quadrant represent instances where TPT was found to be more effective than no TPT in reducing active TB incidence; however, TPT was more expensive than no TPT. Data points in the bottom right quadrant represent instances where TPT was found to be more effective than no TPT in reducing active TB incidence, and TPT was less expensive than no TPT. The lack of data points in the top left and bottom left quadrants means that there was no instance where TPT was predicted to be less effective than no TPT in reducing active TB incidence. The different shapes of data points represent different TPT regimen categories; filled black circles (●) represent INH-based regimens longer than 12 months, filled gray squares (■) represent INH-based regimens between 6 and 12 months, and filled gray triangles (▲) represent rifamycin-based regimens. The dashed line represents an incremental cost-effectiveness value of $1,000 (where incremental cost active TB case averted = $1,000).

Despite this heterogeneity in model parameter values, in all studies, authors concluded TPT was predicted to be cost-effective compared to no TPT, even with diverse willingness-to-pay thresholds specific to each study setting. Four studies found TPT was predicted to be cost saving compared to no TPT [15,28,54,68], and 2 studies concluded TPT was estimated to be "highly" cost-effective [20,30]. Three studies set in HICs with low TB incidence concluded that using TST or IGRA to guide the decision to provide TPT was potentially more cost-effective than providing TPT to all PLHIV [44,46,69]. On the other hand, 2 studies set in LMICs concluded that providing TPT to all pregnant women living with HIV would be potentially more cost-effective than using TST or IGRA to guide TPT decisions in this population [41,43].

As seen in Fig 4, all studies found that providing TPT to PLHIV was predicted to be more effective at averting active TB cases than not providing TPT (Fig 4); a minority (*n* = 4) of these studies found that TPT was potentially both cost saving and more effective than no TPT. Comprehensive outcomes and conclusions from each study are reported in Tables I and J in S1 Text.

## Determinants of study outcomes

The data extracted from studies for the quantitative analyses are summarized in Table K in S1 Text as well as S1 Data.

## Regression analysis for predictors of active TB reduction

According to univariable quantile regression analyses for 95 modeled strategies (i.e., study arms) within studies, model input parameters that were significantly associated with

**Table 2. Univariable regression models: percent reduction in active TB incidence comparing TPT to no TPT.**

| Categorical variables: group of interest | Categorical variables: reference | Number of strategies | Change in percent reduction of active TB incidence | |
| --- | --- | --- | --- | --- |
| | | | Estimate | 95% CI |
| **Strategy is set in an HIC** | **Strategy is set in an LMIC**[*] | **89** | **20.4%** | **5.9% to 29.7%** |
| Strategy includes ART-related variables in analysis | Strategy does not include ART-related variables in analysis | 95 | 0.8% | −22.9% to 10.8% |
| Strategy models rifamycin-based regimen | Strategy models an INH regimen 6 to 12 months[**] | 75 | 0.8% | −14.5% to 27.0% |
| Strategy models an INH regimen >12 months[†] | Strategy models an INH regimen 6 to 12 months[**] | 75 | −0.1% | −13.1% to 7.0% |
| **Continuous variables** | **Definition of unit increase** | **Number of strategies** | Change in percent reduction of active TB incidence | |
| | | | Estimate | 95% CI |
| LTBI prevalence[††] | 10% | 61 | −2.0% | −4.3% to 0.3% |
| Time horizon | 1 year | 94 | 0.2% | −0.1% to 0.6% |
| TPT efficacy in preventing active disease | 10% | 72 | 0.8% | −0.4% to 2.3% |
| Level of TPT adherence | 10% | 30 | 1.5% | −6.8% to 4.6% |
| **Probability of fatal adverse event** | **0.1%** | **26** | **−0.1%** | **−0.4% to −0.0%** |

Values in this table represent the median change in percent reduction of active TB incidence between PLHIV given TPT and PLHIV not given TPT; values were estimated using quantile regression. Variables with CIs on the same side of the null (0) are bolded.

[*] Example interpretation: Studies set in HICs reported a reduction in active TB with TPT that was 20.4% greater than studies set in LMICs. This difference may have been as high as 29.7% more reduction or as little as 5.9% more reduction.

[**] These regimens include 6, 9, and 12 months of INH.

[†] These regimens include 36 months and lifetime INH.

[††] Example interpretation: For every 10% increase in LTBI prevalence, TPT resulted in 2.0% less reduction in active TB, compared to no TPT. This difference may have been as high as 0.3% more reduction or as low as 4.3% less reduction per 10% increase in LTBI prevalence.

ART, antiretroviral therapy; CI, confidence interval; HIC, high-income country; INH, isoniazid; LTBI, latent tuberculosis infection; PLHIV, people living with HIV; TB, tuberculosis; TPT, tuberculosis preventative therapy.

effectiveness of TPT were country income classification and the probability of a fatal adverse event due to TPT (Table 2). Importantly, we define a parameter to be significant if its confidence interval (CI) does not cross the null (0). TPT was more effective in reducing active TB incidence in HICs, compared to LMICs; with 20.4% (95% CI: 5.9% to 29.7%) greater reduction of active TB incidence in HICs. TPT was less effective in reducing active TB incidence as the probability of fatal adverse events increased, with 0.1% (95% CI: 0% to 0.4%) less reduction for every 0.1% increase in the probability of a fatal adverse event due to TPT. In multivariable quantile regression analyses, country income classification and TPT regimen remained significantly associated with relative reduction in active TB incidence (Table 3). In 4 different models including all categorical variables and additionally including various continuous variables, TPT was more effective in reducing active TB incidence in HICs, compared to LMICs. Consistently, models found INH regimens longer than 12 months in duration were most effective at reducing active TB, followed by INH regimens 6 to 12 months in duration and rifamycin-based regimens. This is likely due to modeling methodology; as long as an individual is on TPT, they have a lower probability of progressing to TB disease. As such, longer regimens appear more effective.

## Regression analysis for predictors of incremental net monetary benefit

According to univariable quantile regression analyses for 47 modeled strategies within studies, model input parameters that were significantly associated with cost-effectiveness of TPT were

**Table 3. Multivariable regression models: percent reduction in active TB incidence comparing TPT to no TPT.**

| | | Model 1 (n = 72) | | Model 2 (n = 72) | | Model 3 (n = 72) | | Model 4 (n = 72) | |
|---|---|---|---|---|---|---|---|---|---|
| **Categorical variables: group of interest** | **Categorical variables: reference** | **Change in percent reduction of active TB incidence** | | | | | | | |
| | | Estimate | 95% CI | Estimate | 95% CI | Estimate | 95% CI | Estimate | 95% CI |
| Set in an HIC | Set in an LMIC | **29.4%**[*] | **4.9% to 36.3%** | 17.9% | **4.9% to 37.9%** | 29.8% | **5.9% to 39.3%** | 29.6% | **6.2% to 39.3%** |
| Strategy includes ART-related variables in analysis | Strategy does not include ART-related variables in analysis | −19.6% | −65.3% to 0.6% | −30.8% | −63.4% to 4.6% | −21.5% | −64.8% to 0.2% | −21.3% | −65.9% to 0.9% |
| Strategy models rifamycin-based regimen | Strategy models an INH regimen 6 to 12 months** | −22.1% | **−53.3% to −11.2%** | −33.8% | **−51.1% to −7.9%** | −23.5% | **−52.5% to −7.5%** | −23.7% | **−57.3% to −7.0%** |
| Strategy models an INH regimen >12 months[†] | Strategy models an INH regimen 6 to 12 months** | 7.2% | **2.1% to 13.5%** | 6.4% | **3.9% to 13.7%** | 6.7% | **5.8% to 14.9%** | 7.2% | **5.9% to 15.0%** |

Values in this table represent the median change in percent reduction of active TB incidence between PLHIV given TPT and PLHIV not given TPT; values were estimated using quantile regression. Each column titled "Model" illustrates the results of one multivariable model. Model 1 only included only categorical variables, Model 2 included time horizon in addition, Model 3 included TPT efficacy, and Model 4 included both; missing values for time horizon and TPT efficacy were imputed using medians. Estimates for time horizon and TPT efficacy are not shown as they were negligible. Variables with CIs on the same side of the null (0) are bolded.

* Example interpretation: Controlling for ART use and TPT regimen category, studies set in HICs reported a reduction in active TB with TPT that was 29.4% greater than studies set in LMICs. This difference may have been as high as 36.3% more reduction or as little as 4.9% more reduction.

** These regimens include 6, 9, and 12 months of INH.

† These regimens include 36 months and lifetime INH.

ART, antiretroviral therapy; CI, confidence interval; HIC, high-income country; INH, isoniazid; LMIC, low- and middle-income country; LTBI, latent tuberculosis infection; PLHIV, people living with HIV; TB, tuberculosis; TPT, tuberculosis preventative therapy.

country income classification, consideration of ART use (i.e., whether a parameter for ART efficacy or cost was considered), TPT regimen, time horizon, and TPT efficacy in preventing active disease (Table 4). TPT was more cost-effective in HICs, compared to LMICs (Incremental Net Monetary Benefit [INMB] = $3,566, 95% CI: $210 to $7,575). TPT was less cost-effective among strategies that considered the use of ART (INMB = −$477, 95% CI: −$1,364 to −$173). Moreover, 3 INH RPT was the only rifamycin-based regimen considered in this group of studies and was the most cost-effective TPT regimen, followed by INH regimens 6 to 12 months long, followed by INH regimens longer than 12 months.

In multivariable quantile regression analyses, however, only country income classification and ART use remained significantly associated with incremental net monetary benefit (Table 5). In 4 different models including all categorical variables and additionally including various continuous variables, TPT was more cost-effective in HICs, compared to LMICs. As well, in all 4 models, strategies that considered the use of ART were less cost-effective than strategies that did not consider the use of ART. Studies that considered ART use had lower values for TPT efficacy and shorter time horizons, which may explain why they were negatively associated with cost-effectiveness (Fig A in S1 Text).

## Meta-analysis

Meta-analysis of incremental net monetary benefit of TPT compared to no TPT in LMICs led to a pooled estimate of $271 (95% CI: −$81 to $622; p = 0.12). The pooled estimate for incremental net monetary benefit in HICs was larger, but less precise due to smaller sample size, at $2,568 (95% CI: −$32,115 to $37,251; p = 0.52) (Table 6).

The pooled percent reduction in active TB incidence in HICs was 37% (95% CI: −34% to 100%; p = 0.13). In LMICs, it was 20% (95% CI: 13% to 27%; p < 0.001). We were able to further stratify this subset of strategies by time horizon and TPT regimen. Strategies that had

**Table 4. Univariable regression models: incremental net monetary benefit comparing TPT to no TPT (2020 USD).**

| Categorical variables: group of interest | Categorical variables: reference | Number of strategies | Incremental net monetary benefit | |
|---|---|---|---|---|
| | | | Estimate | 95% CI |
| Strategy is set in an HIC | Strategy is set in an LMIC | 46 | $3,566 | $210 to $7,575 |
| Strategy includes ART-related variables in analysis | Strategy does not include ART-related variables in analysis[*] | 47 | −$477 | −$1,364 to −$173 |
| Strategy models 3 INH RPT | Strategy models an INH regimen 6 to 12 months[**] | 47 | $124 | $53 to $2,997 |
| Strategy models an INH regimen >12 months[†] | Strategy models an INH regimen 6 to 12 months[**] | 47 | −$86 | −$278 to −$33 |
| **Continuous variables** | **Definition of unit increase** | **Number of strategies** | **Incremental net monetary benefit** | |
| | | | Estimate | 95% CI |
| LTBI prevalence | 10% | 32 | $7 | −$249 to $159 |
| **Time horizon** | **1 year** | **47** | **$5** | **$1 to $87** |
| **TPT efficacy in preventing active disease[††]** | **10%** | **41** | **$36** | **$14 to $64** |
| Level of TPT adherence | 10% | 21 | −$440 | −$3,420 to $20 |
| Probability of fatal adverse event | 0.1% | 18 | $4 | −$144 to $20 |

Values in this table represent the median change incremental net monetary benefit between PLHIV given TPT and PLHIV not given TPT; values were estimated using quantile regression. Variables with CIs on the same side of the null (0) are bolded.

[*] Example interpretation: Strategies that do not consider the use of ART find TPT to be more cost-effective compared to no TPT than strategies that do consider the use of ART; on average, the incremental net monetary benefit is $477 higher among studies that do not consider the use of ART, on average. The incremental net monetary benefit could be as high as $1,364 or as low as $173.

[**] These regimens include 6, 9, and 12 months of INH.

[†] These regimens include 36 months and lifetime INH.

[††] Example interpretation: As TPT efficacy increases, the cost-effectiveness of TPT compared to no TPT increases; for every 10% increase in TPT efficacy, the incremental net monetary benefit increases by $36, on average. The incremental net monetary benefit could be as high as $64 or as low as $14.

ART, antiretroviral therapy; CI, confidence interval; HIC, high-income country; INH, isoniazid; LMIC, low- and middle-income country; LTBI, latent tuberculosis infection; PLHIV, people living with HIV; RPT, rifapentine; TPT, tuberculosis preventative therapy; USD, United States dollar.

longer time horizons had a higher pooled estimate for percent reduction in active TB incidence than studies with shorter time horizons. Similarly, strategies that considered INH regimens longer than 12 months had a higher pooled estimate for percent reduction in active TB incidence than strategies that considered INH regimens 6 to 12 months long (Fig 5). There were no strategies among this subset that considered rifamycin-based regimens. We were limited by the number of strategies, and, therefore, could not consider more stratifications.

## Sensitivity analysis

Repeating pooling analyses with a lower (0.5× GDP) and higher (3× GDP) willingness-to-pay threshold demonstrated that incremental net monetary benefit is proportional to scale; as GDP increased or decreased, the pooled incremental net monetary benefit also increased or decreased. Similarly, repeating univariable and multivariable analyses with a lower and higher willingness-to-pay threshold did not change any conclusions; variables that were significantly associated with incremental net monetary benefit tended to remain significantly associated (see Tables N–Q in S1 Text). Additional results are articulated in Tables L–N and Figs D–K in S1 Text.

## Discussion

Despite variability in determinants that may affect the cost-effectiveness of TPT, all 61 studies included in this review concluded that TPT was predicted to be effective and/or cost-effective

**Table 5. Multivariable quantile regression models: incremental net monetary benefit comparing TPT to no TPT (2020 USD).**

| Categorical variables: group of interest | Categorical variables: reference | Model 1 ($n = 46$) | | Model 2 ($n = 46$) | | Model 3 ($n = 46$) | | Model 4 ($n = 46$) | |
|---|---|---|---|---|---|---|---|---|---|
| | | Incremental net monetary benefit | | | | | | | |
| | | Estimate | 95% CI | Estimate | 95% CI | Estimate | 95% CI | Estimate | 95% CI |
| Set in an HIC | Set in an LMIC | **$3,539** | **$956 to $42,990** | **$3,453** | **$1,088 to $40,429** | **$3,472** | **$918 to $42,982** | **$3,686** | **$918 to $42,985** |
| Strategy includes ART-related variables in analysis | Strategy does not include ART-related variables in analysis | **−$612**[*] | **−$1,385 to −$437** | **−$651** | **−$1,420 to −$427** | **−$593** | **−$1,355 to −$279** | **−$456** | **−$1,255 to −$348** |
| Strategy models rifamycin-based regimen | Strategy models an INH regimen 6 to 12 months[**] | −$531 | −$18,423 to $1,238 | −$514 | −$7,666 to $437 | −$467 | −$17,173 to $476 | −$351 | −$7,738 to $275 |
| Strategy models an INH regimen >12 months[†] | Strategy models an INH regimen 6 to 12 months[**] | −$17 | −$88 to $69 | −$38 | −$74 to $66 | −$46 | −$187 to $50 | −$58 | −$221 to $30 |

Values in this table represent the median change in incremental net monetary benefit between PLHIV given TPT and PLHIV not given TPT; values were estimated using quantile regression. Each column titled "Model" illustrates the results of one multivariable model. Model 1 only included only categorical variables, Model 2 included time horizon in addition, Model 3 included TPT efficacy, and Model 4 included both; missing values for time horizon and TPT efficacy were imputed using medians. Estimates for time horizon and TPT efficacy are not shown as they were negligible. Variables with CIs on the same side of the null (0) are bolded.

[*] Example interpretation: Controlling for country income level and TPT regimen category, strategies that do not consider the use of ART find TPT to be more cost-effective compared to no TPT than strategies that do consider the use of ART; on average, the incremental net monetary benefit is $612 higher among studies that do not consider the use of ART, on average. This incremental net monetary benefit could be as high as $1,385 or as low as $437.

[**] These regimens include 6, 9, and 12 months of INH.

[†] These regimens include 36 months and lifetime INH.

ART, antiretroviral therapy; CI, confidence interval; HIC, high-income country; INH, isoniazid; LMIC, low- and middle-income country; PLHIV, people living with HIV; TPT, tuberculosis preventative therapy; USD, United States dollar.

**Table 6. Pooled incremental net monetary benefit (2020 USD) and percent reduction in active TB incidence comparing TPT to no TPT.**

| | Value | | | $\tau$ | | $I^2$ | |
|---|---|---|---|---|---|---|---|
| | Estimate | 95% CI | p-value | Estimate | 95% CI | Estimate | 95% CI |
| **LMICs** | | | | | | | |
| Pooled incremental net monetary benefit ($n = 10$) | $271 | −$81 to $622 | 0.12 | $441 | $164 to $909 | 67% | 35% to 83% |
| Pooled percent reduction in active TB Incidence ($n = 23$) | 20% | 13% to 27% | <0.001 | 15% | 11% to 21% | 96% | 95% to 97% |
| Among strategies that have a time horizon <5 years ($n = 3$) | 10% | −22% to 43% | 0.42 | 11% | 5% to 34% | 96% | 91% to 98% |
| Among strategies that have a time horizon ≥5 years ($n = 20$) | 21% | 13% to 28% | <0.001 | 15% | 11% to 24% | 96% | 95% to 97% |
| Among strategies that include an INH regimen 6 to 12 months long ($n = 13$)[*] | 17% | 8% to 26% | 0.001 | 14% | 9% to 23% | 95% | 93% to 97% |
| Among strategies that include an INH regimen longer than 12 months ($n = 6$)[*] | 24% | 3% to 45% | 0.04 | 20% | 8% to 55% | 86% | 73% to 93% |
| **HICs** | | | | | | | |
| Pooled incremental net monetary benefit ($n = 2$) | $2,568 | −$32,115 to $37,251 | 0.52 | $2,122 | NA | 0% | NA |
| Pooled percent reduction in active TB incidence ($n = 3$) | 37% | −34% to 100% | 0.13 | 25% | 12% to 82% | 95% | 88% to 98% |

$\tau$ = square root of between study variance; $I^2$ = measure of heterogeneity.

[*] There are only 19 strategies that were eligible for stratification by TPT regimen category. The other 4 strategies did not report duration of INH.

CI, confidence interval; HIC, high-income country; INH, isoniazid; LMIC, low- and middle-income country; TB, tuberculosis; TPT, tuberculosis preventive treatment; USD, United States dollar.

Pooling Percent Reduction in Active TB for LMICs by TPT Regimen Category

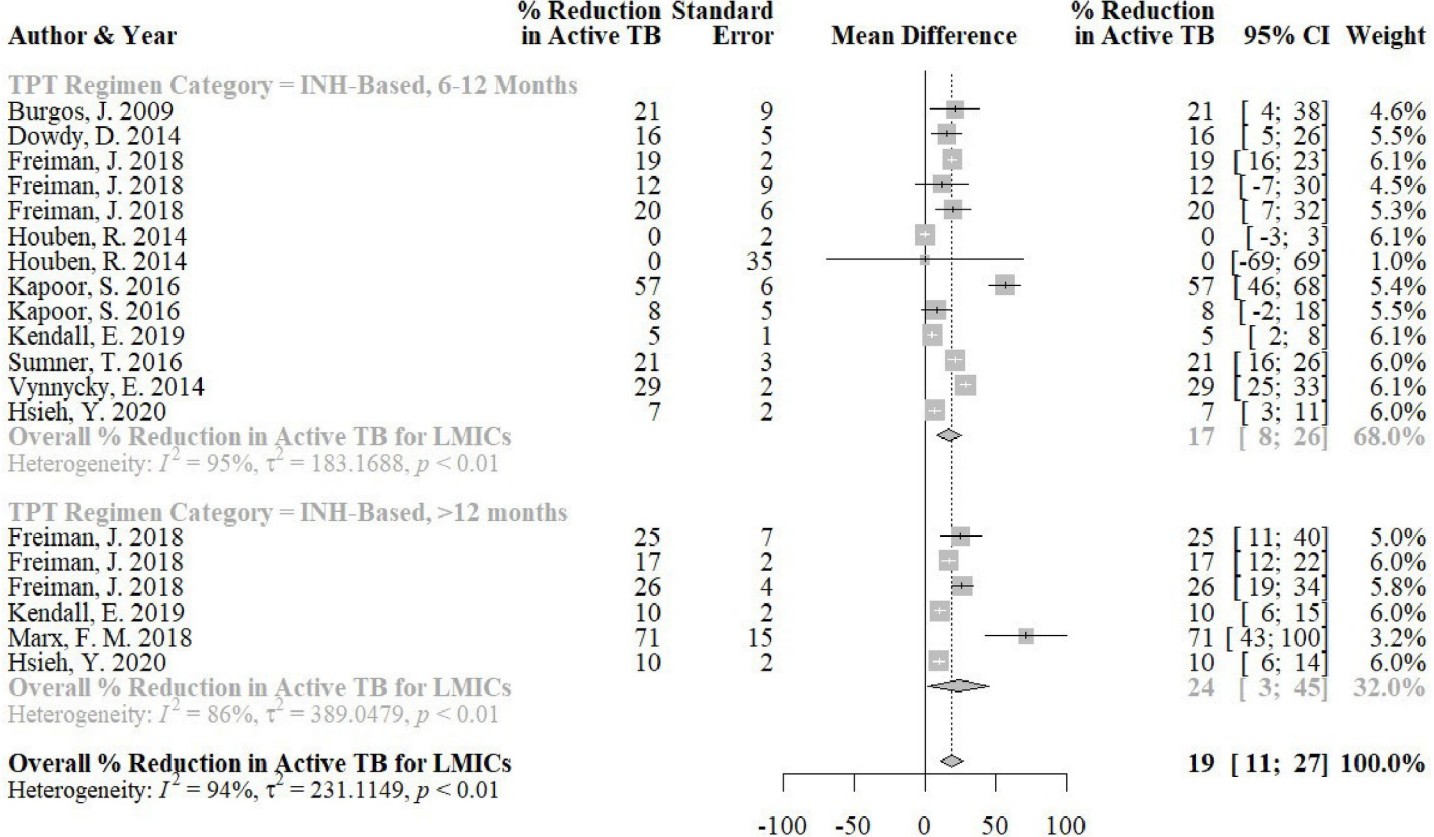

Pooling Percent Reduction in Active TB for HICs, all TPT Regimens

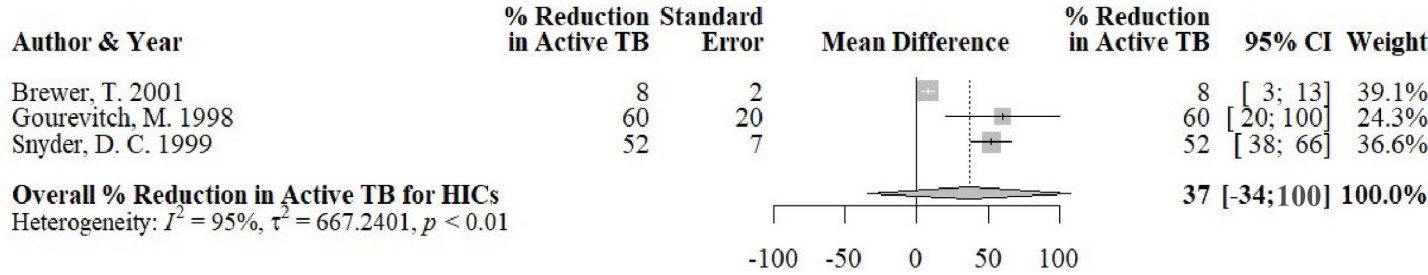

**Fig 5. Forest plot: percent reduction in active TB in studies comparing TPT to no TPT, by groups of regimens and country income level.** CI, confidence interval; HIC, high-income country; INH, isoniazid; LMIC, low- and middle-income country; RIF, rifampin; TB, tuberculosis; TPT, tuberculosis preventive therapy. No subgroup analyses done for HICs to small number of studies. No RIF-based regimens included in LMICs that had sufficient information for pooling.

—and, sometimes, cost saving—regardless of regimen and study setting. This was supported by the pooled incremental net monetary benefit being positive in all settings. We identified several potential determinants of cost-effectiveness in modeling studies; however, only country-level income and consideration of ART use or cost remained associated with TPT cost-effectiveness in multivariable analysis.

The universal conclusion that providing TPT to PLHIV is cost-effective may support initiatives to further expand provision of TPT to PLHIV. In general, cost and cost-effectiveness are

key components of program scalability [73]. Studies that model the potential cost, effectiveness, and cost-effectiveness of TPT may offer guidance for resource allocation during program scale-up [5]. Although modeling studies are powerful in this regard, it is important to consider modeling assumptions prior to enacting any decision, as evidenced by the widely varying assumptions and input parameters considered in the studies in this review [74].

Other important findings are that TPT's apparent effectiveness was greatest among studies that considered INH-based regimens longer than 12 months. This is likely due to how regimen length was considered in models; generally, as long as an individual was given TPT, they had a lower or negligible probability of progressing to active TB. As such, a longer regimen would mean a longer time with limited progression to disease. Our regression analyses seem to support this, with the shortest regimens (rifamycin based) being less cost-effective, although this was not statistically significant. This is likely due to 3 INH RPT being the most common rifamycin-based regimen and the high costs of rifapentine when it was first introduced. Costs of rifapentine have since fallen, and one study suggested that a lower cost of rifapentine was associated with substantial gains in cost-effectiveness [39].

A strength of this review was the inclusion of a large number of studies, which assessed different TPT regimens in many different settings. Interestingly, although values of input parameters varied widely, parameters that were based on published data were not significantly different from parameters that were based on assumptions. The heterogeneity of input parameters enabled us to quantitatively assess the impact of differences in these key determinants on effectiveness and cost-effectiveness in modeling studies. Importantly, despite the variability in input parameters and methods, the conclusions were the same—TPT is predicted to be effective and cost-effective in all settings and with all regimens considered. Hence, this conclusion can be considered very robust.

However, the substantial heterogeneity of input parameter values and assumptions did make these studies difficult to summarize and limited the extent of pooling through meta-analytical techniques. Other systematic reviews of cost-effectiveness or dynamic modeling studies have also concluded that there is very substantial variability in study methodology and parameterization [75–83]. Two of these reviews concluded that these inconsistencies limited inferences [78,80]. The heterogeneity of model inputs emphasizes the need for better standardization of models for TB, exemplified by a published "modeler's wish list" [84]. For example, model input parameters, such as rate of progression after recent infection and reactivation after remote infection, are often taken from studies that precede the advent of antiretroviral therapy, and adherence/completion parameters generally do not consider shorter TPT regimens [84]. This contributes to heterogeneity in model input parameters. Another common methodological issue was inconsistency in reporting uncertainty in model estimates. In particular, this inconsistency limited the ability to meta-analyze studies.

Another limitation could be publication bias toward positive effects of TPT—while clinical trials have consistently shown TPT is effective, models showing the opposite may not be published. Individual level adherence to TPT is likely to vary among a population, but model parameterization did not allow us to investigate this factor. Finally, our bibliographic databases favor English publications, which may introduce bias [85].

Conducting this review also highlighted areas that require further research. This includes examining costs as well as effectiveness of shorter rifamycin-based TPT regimens, costs, and effectiveness in specific subpopulations with HIV, such as pregnant women and injection drug users, and considering a broader societal perspective, rather than simply the health system. Understanding the nuances in costs and impacts of TPT among vulnerable populations will allow for effective program delivery that addresses challenges and barriers unique to those populations [40]. In addition, shorter, rifamycin-based regimens are increasingly

recommended for TPT worldwide, making it important to understand their potential costs, effectiveness, and cost-effectiveness, compared to other treatment options [86].

In sum, our review found that there is great heterogeneity in methodology, parameterization, and assumptions between studies that modeled the costs, effectiveness, and cost-effectiveness of TPT among PLHIV. Despite these inconsistencies, all studies reviewed concluded that providing TPT to PLHIV was potentially effective and cost-effective compared to not providing TPT. This supports greater resource allocation in all settings to expand programs that deliver TPT to PLHIV.

## Supporting information

**S1 PRISMA Checklist. Checklist.** PRISMA, Preferred Reporting Items for Systematic Reviews and Meta-Analyses.
(PDF)

**S1 Data. The data extracted from studies for the quantitative analyses are summarized in this file as well as in Table F in S1 Text.**
(XLSX)

**S1 Text. Table A:** Search strategy for MEDLINE, Embase, and Web of Science. **Table B:** Quality assessment checklist. **Table C:** Quality assessment results. **Table D:** Number of studies that used each type of data source for key input parameter categories. **Table E:** Comparing input parameters that were based on data to those that were based on assumptions. **Table F:** description of included studies. **Table G:** Key outcomes among studies that report effectiveness or utility outcomes only. **Table H:** Key outcomes and results of studies that reported costs and cost-effectiveness outcomes (2020 USD). **Table I:** Detailed outcomes of studies that reported cost and cost-effectiveness outcomes. **Table J:** Detailed outcomes of studies that reported effectiveness outcomes only. **Table K:** Data used for regression analyses. **Table L:** Comparing one-way sensitivity analysis results across cost and cost-effectiveness studies. **Table M:** Comparing one-way sensitivity analysis results across studies that only report effectiveness or utility outcomes. **Table N:** Threshold analysis results among included studies (that reported key thresholds where conclusions changed). **Table O:** Effect of 0.5× and 3× GDP per capita willingness-to-pay threshold on univariable analysis of incremental net monetary benefit. **Table P:** Effect of 0.5× and 3× GDP per capita willingness-to-pay threshold on multivariable analysis of incremental net monetary benefit. **Table Q:** Effect of 0.5× and 3× GDP per capita willingness-to-pay threshold on pooling analysis of incremental net monetary benefit. **Fig A:** Comparing the association between art use, TPT efficacy, and time horizon (model inputs). **Fig B:** Forest plot: pooling incremental net monetary benefit in LMICs. **Fig C:** Forest plot: pooling incremental net monetary benefit in HICs. **Fig D:** Model inputs: comparing time horizon by TPT regimen category and country-level income. **Fig E:** Model inputs: comparing TPT efficacy in preventing active TB by TPT regimen category and country-level income. **Fig F.** Model inputs: comparing level of TPT adherence by TPT regimen category and country-level income. **Fig G:** Model outputs: comparing per-person cost of strategies that included TPT by TPT regimen category and country-level income. **Fig H:** Model inputs: select variables and their relationship to the per-person cost of strategies that included TPT. **Fig I:** Model inputs versus model outputs: comparing calculated effectiveness based on model inputs (efficacy × adherence) to reported effectiveness based on model outputs (percent reduction in active TB incidence). **Fig J:** Model outputs: comparing reduction in active TB incidence by TPT regimen category and country-level income. **Fig K:** Model outputs: comparing incremental cost per active TB case averted by country-level income and TPT regimen category. **Fig R:**

Data extraction form. GDP, gross domestic product; HIC, high-income country; LMIC, low- and middle-income country; TB, tuberculosis; TPT, tuberculosis preventive treatment; USD, United States dollar.
(DOCX)

## Author Contributions

**Conceptualization:** Jonathon R. Campbell, Olivia Oxlade, Dick Menzies.

**Data curation:** Aashna Uppal, Samiha Rahman.

**Formal analysis:** Aashna Uppal, Samiha Rahman, Jonathon R. Campbell, Olivia Oxlade, Dick Menzies.

**Funding acquisition:** Olivia Oxlade, Dick Menzies.

**Investigation:** Aashna Uppal, Samiha Rahman, Jonathon R. Campbell, Olivia Oxlade, Dick Menzies.

**Methodology:** Aashna Uppal, Samiha Rahman, Jonathon R. Campbell, Olivia Oxlade, Dick Menzies.

**Software:** Aashna Uppal, Samiha Rahman.

**Supervision:** Jonathon R. Campbell, Olivia Oxlade, Dick Menzies.

**Validation:** Aashna Uppal, Samiha Rahman, Jonathon R. Campbell, Olivia Oxlade, Dick Menzies.

**Visualization:** Aashna Uppal, Samiha Rahman.

**Writing – original draft:** Aashna Uppal, Samiha Rahman.

**Writing – review & editing:** Aashna Uppal, Samiha Rahman, Jonathon R. Campbell, Olivia Oxlade, Dick Menzies.

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
