## [Editor Report · Decision Letter 0]

14 Dec 2020

Dear Dr Menzies, 

Thank you for submitting your manuscript entitled "Economic and modelling evidence for tuberculosis preventive therapy among people living with HIV: a systematic review" for consideration by PLOS Medicine.

Your manuscript has now been evaluated by the PLOS Medicine editorial staff and I am writing to let you know that we would like to send your submission out for external assessment.

Please resubmit your paper as a research article; and remove the attached Collections form. 

Once your full submission is complete, your paper will undergo a series of checks in preparation for assessment.

Kind regards,

Richard Turner, PhD

Senior editor, PLOS Medicine

rturner@plos.org

---

## [Decision Letter · Decision Letter 1]

17 Feb 2021

Dear Dr. Menzies,

Thank you very much for submitting your manuscript "Economic and modelling evidence for tuberculosis preventive therapy among people living with HIV: a systematic review" (PMEDICINE-D-20-06011R1) for consideration at PLOS Medicine. 

Your paper was discussed among the editors and sent to independent reviewers, including a statistical reviewer. The reviews are appended at the bottom of this email and any accompanying reviewer attachments can be seen via the link below:

[LINK]

In light of these reviews, we will not be able to accept the manuscript for publication in the journal in its current form, but we would like to invite you to submit a revised version that addresses the reviewers' and editors' comments fully. You will recognize that we cannot make a decision about publication until we have seen the revised manuscript and your response, and we expect to seek re-review by one or more of the reviewers. 

We hope to receive your revised manuscript by Mar 10 2021 11:59PM. Please email us (plosmedicine@plos.org) if you have any questions or concerns.

Please let me know if you have any questions. We look forward to receiving your revised manuscript in due course. 

Sincerely,

Richard Turner, PhD

rturner@plos.org

Did this study have a protocol or prespecified analysis plan, and was it registered prior to being carried out?

Please update the search to the end of 2020, say. 

Please remove the information on funding from the title page. In the event of publication, this information will appear in the article metadata, via entries in the submission form. 

Please make that "data were" in the abstract; just before that, a duplicate full point needs to be removed. 

We suggest adding an additional sentence, say, to the abstract to summarize study designs, settings and other study characteristics such as regimens.

Please add a new final sentence to the "Methods and findings" subsection of your abstract, which should quote 2-3 of the study's main limitations. 

After the abstract, please add a new, accessible "author summary" section in non-identical prose. You may find it helpful to consult one or two recent research articles published in PLOS Medicine to get a sense of the preferred style. 

Please add a completed checklist for the most appropriate reporting guideline as a supplementary document, labelled "S1_PRISMA_Checklist" or similar, referred to in your Methods section. In the checklist, individual items should be referred to by section (e.g., "Methods") and paragraph number rather than by line or page numbers, as the latter generally change in the event of publication. 

Please make that "low- and middle-income countries" throughout. 

Please ensure that journal names are abbreviated consistently in your reference list. Please add "U S A" to the journal name for reference 18 and any other relevant entries. 

Comments from the reviewers:

*** Reviewer #1: 

The authors conducted a systematic review on economic and modeling evidence for TB preventive therapy in PLHIV. This is an important area of research as TPT in both the general and PLHIV populations is an important early intervention for TB control and prevention. As with many past systematic review studies carried out by the authors' TB research group, this study carries forward the same basic strengths in thoroughness and principle-based systematic assessment of the literature in the research topic. However, this reviewer finds that this manuscript lacks clear descriptions in some of the analytic methods used and in-depth discussions on key issues in the heterogeneity that exists across the studies. Compared to the complexities of the topics that the authors investigated in this study, this reviewer finds the author's approach to writing the manuscript have been a bit to concise, which makes some of the study's findings difficult to follow (i.e. the richness of the information captured in this study are not adequately reflected in the writing). Therefore, this reviewer has following two key improvements to enhance both the interpretation and the content of the manuscript: 

1) The authors state in their objective that this study was carried out to "systematically review" and "synthesize the costs, cost-effectiveness, as well as risks and effect on TB morbidity and mortality associated with TPT provided to PLHIV." This reviewer finds that in general, this manuscript leaves it up to the readers to synthesize many of the important findings (e.g. the highly heterogenous nature of data, methods, and outcomes across the study) and understand what need to improve this issue for future studies. These can be improved by looking at study level differences (e.g. modeling techniques, costing methods, validity of parameter values) within a larger groups of studies (e.g. model-based studies: decision analysis vs. transmission) as this would be very difficult to do across all studies (hence the highly heterogenous nature of study methods and use of parameters)

2) Costs are listed as one of the key topic areas that were considered for the main objective. However, the authors limit their assessment on this at high-level discussion. Given that costs parameters and estimates are equal component in cost-effectiveness analyses (and the sole objective of cost analyses studies), it will be important to dedicate a section within this manuscript to describe how studies differ on their methods in evaluating (and or the use of) costs of TPT (most important) and other health services included in their assessment (e.g. top down vs. bottom-up, or use of more empiric approach vs. referenced or assumed cost estimates) 

Also, following are section and line-specific comments which should guide the authors in addressing the two issues raised above:

A. Introduction 

Line 80-81: Transmission modeling and cost-effectiveness studies only cover part of the key evidence that are needed to improve TPT coverage and service delivery. Please add 

B. Methods

Line 95-96: Can authors expand on what types of economic evaluations were considered (e.g. cost analysis, budget impact analyses, etc.) 

Line 111-112: What does the author mean by "at least one key input parameters"? If study provided appropriate source for one parameter but used assumptions for rest, would this study deem to make the mark for quality? Also, how did the authors determine appropriateness of the source? 

Line 136-137: also costs are incremental (i.e. provision of TPT would increase costs compared to the status quo)

Line 140-143: Can authors provide more descriptions as to why forward selection was chosen over other methods (e.g. all-in vs. backwards elimination vs. bidirectional elimination vs. score comparison)? This reviewer is not asking to perform and compare these model selection methods but is asking the authors provide one to two short statements on the rationale for choosing forward selection method (for the benefit of the readers). 

General comment regarding description of the regression method: This reviewer feels that the authors' minimalistic description of the regression analyses that were performed for the study makes it difficult to connect the dots across data and information presented in Tables S8, 3a, b, & c, and the texts in the methods section. In short, readers are having to juggle across various parts of the manuscript to truly understand what was exactly done (e.g. there were three main regression outputs assessed study level determinants on 1. reduction in active TB; 2. Reduction in TB-related mortality; and 3. ICER estimates for TPT (vs. no TPT). 

Following is an example of the disconnect this reviewer experiencing when reading through the text/tables/figures: It's clear to this reviewer than the regression output is either change in percent reduction of mortality (vs. reference standard) or ICER (measured as cost per active TB averted), but it's unclear how number of strategies were used in the regression analyses. 

Suggestion: Can authors provide a bit more description on how the regression analysis was conducted? For lay readers, it would be helpful to spell out the equation and/or text to enhance the understanding of how data in Table S8 was assessed to provide outputs in shown in Table 3a, b & c. The authors provide example interpretations as sub-texts for Tables 3a, b, and c, but this reviewer feels that the authors should provide more information about their regression analysis in the methods section. 

C. Results 

Line 146-148: In the abstract, the authors write "search identified 6,135 titles" (vs. 4,228 unique records in this line). It'd be important for the authors to be consistent in search results and selection process figures in the respective sections of the manuscript. 

Line 156: For the rest of 30 studies, what methods did they use for the analysis and reporting of the economic evaluation of TPT in PLHIV? The authors break this down in Table 1 (modeling studies only), but for the text, it'd better to report as the following, for example: "50 studies (out of 57) used modeling methods to evaluate impact and/or cost-effectiveness TPT in PLHIV; 27 used decision analysis method and 23 used transmission modeling. Seven studies that did not use modeling used …(here, describe what type(s) of analysis was done for these studies that did not use modeling methods)." 

Also to this reviewer, decision analysis models can also incorporate transmission model (as developed by one of the co-authors). As such, it may be better to describe these differences as modeling analysis that includes vs. excludes transmission or TB natural history models. 

Analytic horizons: Can the authors classify studies by following categories? 

* No analytic horizons stated or less than 2 years

* Analytic horizon between 2 and 10 years 

* Analytic horizon more than 10 years (for this, indicate in the sub-text maximum analytic horizon used)

Comments on Table 1

* Population � it may be better to report each study's main population investigated and whether PLHIV was a sub-set or main population that was studied. 

* It will be important to further classify studies if they did more detailed empiric costing or simple costing using reported unit cost estimates from other studies (or databases). Also for empiric costing, it'd be good to indicate if studies include costs associated with implementation process (e.g. Sohn H, Tucker A, Ferguson O, Gomes I, Dowdy D. Costing the implementation of public health interventions in resource-limited settings: a conceptual framework. Implement Sci [Internet]. 2020 Dec 29;15(1):86. Available from: https://implementationscience.biomedcentral.com/articles/10.1186/s13012-020-01047-2)

* For studies that modeled or assessed LTBI tests (e.g. either TST or IGRA), were any of these studies compared impact/effectiveness/cost-effectiveness of TST vs. IGRA and how did studies used parameters for diagnostic accuracy for LTBI tests? 

Comments on Table S5

* For probabilities, it'd be important to indicate of they are life time probabilities or time-associated probabilities. 

* Cost of adverse event � It'd be good to indicate what were the most frequently cited adverse event and their associated probability (describe in the sub-title)

Line 209-210 & 255: In the method section or directly next to the text in line 255, can authors provide rational for using $1,000 as a threshold (if possible, provide what would $1,000 per percent reduction in active TB incidence would measure against more traditional ICER estimate - e.g. cost per DALY averted/life years save, TB cases averted etc.)? 

Line 290-291: Was this due to increase in cost or reduced effectiveness? 

D. Discussions (comments are specific to paragraphs)

Potential determinants of CE would also depend on what WTP is considered. These vary significantly from one setting to another. Therefore, I agree with the use of cautionary vocabulary of "potential" but it would be good to indicate under what threshold considerations this study considered these variables to be 'potentially' influencial in determining CE of TPT in PLHIV. 

Third paragraph: It's quite a lengthy discussion on heterogeneity. What is noted here is important and adequate, but other studies have already summarized these issues. What would be beneficial for the readers is to know what types of variabilities/heterogeneities are there across the studies on three key things: 1) input parameters (a. cost ; b. key epidemiologic and TB natural history parameters) 2) modeling strategy; and 3) outcome measures. This study has gone lengths to discover and summarize a lot of information from various studies that investigated impact/effectiveness and cost effectiveness of TPT in PLHIV, but did not adequately described differences across these studies in a systematic manner that could help with better alignment and standardization in evaluating effectiveness/impact and cost-effectiveness of TPT. 

Fourth paragraph: It is important for these modeling studies to separately and conjointly investigate the main drivers of costs, effectiveness, and cost-effectiveness. Many economic evaluation studies that investigate the cost-effectiveness solely focus on ICER estimate. The authors mention that provision of ART modify the effect of TPT among PLHIV. Contrastingly, provision of TPT enables positive health outcomes (i.e. people are surviving longer) which subsequently increases health systems costs. Without investigating these in more detail within the model and reporting these results, many of the future studies investigating the value of TPT will likely suffer the same issue of limited interpretation of the ICERs. Authors can help strengthen the contents of this paragraph by providing a bit more guidance on what are important analyses these modeling studies concerning TPT for PLHIV (or knowledge gaps that can improve overall modeling). 

If the authors consider the fifth paragraph is the most important part of their study ('important finding'), it should be one of the first ones to come. I'd suggest a bit of restructuring of the paragraphs so that ordering of the contents be: 

1) Overall summary of the finidngs

2) Most important finding of the study - TPT in PLHIV is cost-effective

3) Discussion of other key findings (see further comment on this matter below)

4) Discussion on strengths of the author's work

5) Discussion on limitations of the authors work

6) Conclusion

F. Minor content and formatting issues

Line 38: There is an extra dot after the word "quality". 

Line 45: Effectiveness measured as what? 

Line 46: Specify what downstream costs considered (e.g. post TPT health systems and/or patient costs?)

Line 71: Perhaps revise to: "those who may be infected with and are at risk of" 

Line 74-75: Recommend revising the sentence to "Uptake of TPT has increased substantially in recent years, with close to two-fold (1.8 million to 3.6 million between 2018 and 2019) increase in TPT initiation amongst PLHIV." 

Line 75: Can authors provide reference or comment on whether this 50% TPT coverage in PLHIV represent 

Line 92: Please replace "excluded" to "ruled-out"

Line 93-94: Please revise the sentence "study's projected outcomes…" to "study investigated costs, cost-effectiveness (assessed for programmatic yields such as case detection or extended to utility outcomes such as DALYs or QALYs), and epidemiologic impact estimates such as TB incidence, deaths." 

Table 2a & b: Would be better to switch the order of the columns (suggestion: switch the position of the "key outcome reported" with "Comparison Made" so that "Key outcome reported" is next to "Outcome value"). This would better help with interpretation of the units of outcome value. 

*** Reviewer #2: 

SUMMARY

This well-written manuscript summarizes cost-effectiveness and modelling research that has investigated the consequences of preventive treatment of LTBI among individuals with HIV. I think the subject being summarized is worth addressing, but I have some methodological concerns, particularly regarding the regression analysis applied to the cost-effectiveness results.

MAJOR ISSUES

* Page 4, search strategy: I think a critical inclusion criterion would be that the study represented a comparison of two populations/cohorts where the ONLY difference was the receipt vs. non-receipt of IPT by HIV-positive individuals. This is implicit in the title and introduction of the paper, but I don't see this described explicitly at the top of the methods. 

* A more specific concern regarding inclusion/exclusion criteria: I expect some studies described the results of IPT interventions where there is an initial screen for TB disease, and where this screen does not occur in the 'no-IPT' branch. Therefore, the CEA results would represent the costs/benefits of IPT, plus the costs/benefits of TB treatment triggered by the TB disease screening. The first inclusion criterion ('study population included at least a subset of PLHIV in whom active TB had been excluded') could be interpreted as ruling out studies that looked at this typical IPT strategy, but I am not sure if this is what was actually done. If both types of studies were included (ie studies that considered the costs and benefits of TB disease identified by the initial screen), then it would be good to include this as a stratification in the subsequent results.

* The study seems to include papers that compare 'IPT' vs 'no-IPT' scenarios, as well as 'IPT for all' vs. 'IPT only for TST/IGRA positive'. These are quite different comparisons - I think it is fine for them to be considered in the same paper, but these results should not be included in the same quantitative analysis (as in Figure 3?), and care should be taken so that readers do not conflate them.

* Page 6 lines 131: I have concerns about the approach taken to conduct quantitative analyses of the ICER results. By confirming that the ICER denominators are strictly positive (ie IPT strategy always has positive health benefits vs. no-IPT) the authors avoid the stickiest scenario, but this doesn't resolve the issue that the outcome (ICER) is inversely related to the denominator, so very small denominators could lead to very large ICERs. Bottom-coding the ICERs at zero also makes the interpretation of the regression results more difficult (as the denominators are strictly positive, the interpretation of these ICERs as cost-savings seems pretty straightforward), and it is unclear why the arithmetic means that are being calculated through the regression are meaningful quantities. Several of these concerns go away if one considers the median (since not outliers don't matter), and so median regression might represent a more defensible approach. Additional discussion of the difficulties of systematic reviews of cost-effectiveness results are found here (Shields and Elvidge Systematic Reviews 2020 https://systematicreviewsjournal.biomedcentral.com/articles/10.1186/s13643-020-01536-x). On net, I believe the methodological shortcomings one needs to accept to include this kind of analysis are not worth it. I don't think this represents mistakes made by the investigators in implementing the approach, but the problematic conceptual justification of such comparisons.

* As a separate concern about the regression analysis, I am not sure why variable selection is performed. If a variable is of interest I would think it would be included, and predictive performance isn't really an objective here.

* The consideration of costing perspective seems insufficient. Whether the costs of averted TB disease care are included, and whether the costs of additional HIV care are included (a result of improved survival), are both factors that could have a meaningful impact on the ICER, yet these distinctions do not appear in the paper (unless I missed them!).

* Page 13, lines 209: it is stated that 'In all studies, TPT was found to be cost-effective compared to no TPT, even with diverse willingness-to-pay thresholds specific to each study setting'. I don't see where this statement is supported in the results. For something to be cost-effective there needs to be either comparison to an explicit CE threshold, OR direct comparison to other interventions being considered for funding, in addition to a budget limit. Is this a result based on the conclusions drawn by the authors in each paper? If so it would be good to state this. If it is a conclusion drawn based on the extracted ICERs this needs to be done carefully, noting that the historical approach of comparing to 1x and 3x per capita GDP is now increasingly questioned (as an example see Woods et al Value in Health 2016 https://pubmed.ncbi.nlm.nih.gov/27987642/).

MINOR ISSUES

* Page 5, line 120: there is a list of values extracted from the papers, termed 'input parameters'. These are not all input parameters for the models, so perhaps a more general description can be used. In this list, it is unclear what 'consideration of ART use' means.

* Page 11, Table 2a: It is unclear what the 'Outcome Value' is.

* Page 25, lines 333-334: would be important to cite the work that has compared the parameterization and structure of models for LTBI (Ragonnet et al Epidemics 2017; https://www.ncbi.nlm.nih.gov/pmc/articles/PMC6070419/ Menzies et al Lancet Infect Dis 2018 https://pubmed.ncbi.nlm.nih.gov/29653698/; Sumner and White BMC Infect Dis 2020 https://pubmed.ncbi.nlm.nih.gov/33228580/)

*** Reviewer #3: 

[See attachment]

Michael Dewey

***

[LINK]

---

## [Decision Letter · Decision Letter 2]

10 Jun 2021

Dear Dr. Menzies,

Thank you very much for re-submitting your manuscript "Economic and modelling evidence for tuberculosis preventive therapy among people living with HIV: a systematic review & meta-analysis" (PMEDICINE-D-20-06011R2) for consideration at PLOS Medicine.

I have discussed the paper with editorial colleagues and our academic editor and it was also seen again by three reviewers. I am pleased to tell you that, once the remaining editorial and production issues are fully dealt with, we expect to be able to accept the paper for publication in the journal.

[LINK]

Please let me know if you have any questions, and we look forward to receiving the revised manuscript.   

Sincerely,

Richard Turner, PhD

rturner@plos.org

Requests from Editors:

Throughout the paper, please modify the text to soften language such as "was effective ...", which at PLOS Medicine we generally reserve for evidence from randomized studies.

At line 48, please amend the wording to "... was predicted to be effective at averting TB disease." or similar, given the study designs included.

At line 51, please amend the text to "... were estimated to be less than $1500 ...".

Please restructure the author summary so that each of the three subsections contains 2-4 points. For example, the number of studies (61) does not need a specific point and can be included in another. 

Please use the style "... 2 studies" throughout the text, although numbers should be spelt out at the start of sentences (as at present). 

Please quote p values alongside 95% CI, where available.

Please remove the information on funding from the end of the main text. In the event of publication, this information will appear in the article metadata, via entries in the submission form.

Please make that "-$74 to $66" in table 3b.

We generally ask that exact p values are quoted, or "p<0.001"; and note that "p<0.01" appears in fig. 5.

Please ensure that appropriate journal name abbreviations are used, e.g., "Proc Natl Acad Sci U S A" for reference 2 and others; and "PLoS ONE".

Noting reference 12 and others, please list a maximum of 6 author names, followed by "et al.".

Noting reference 37, please ensure that all references have full access information.

Please rename the attached checklist "S1_PRISMA_Checklist" or similar and refer to it by this label around line 125.

Comments from Reviewers:

*** Reviewer #1: 

Authors of the manuscript have adequately address all of the issues raised in my review. Thank you for this important work! 

*** Reviewer #2: 

These revisions largely resolve my original concerns. I just have some minor issues with the NMB analysis.

For the analysis of NMB, there is great uncertainty as to what the correct threshold is for a given country. For this reason it would be helpful to do a sensitivity analysis with alternative values (eg 0.5x and 3x per capita GDP), which could go in the appendix. 

Secondly, the text that describes the NMB calculation needs to be clearer. What was multiplied by per capita GDP to value the health effects? Conventionally this would be DALYs (at least in high-burden settings), and this is the outcome for which conventions around defining WTP as some multiple of per capita GDP are based. Also, the text should state that these NMB is calculated using the per-person costs and DALYs -- while the ICER in unchanged by scale, NMB is proportional to scale, so need to specify this. Also useful to provide some citation for NMB (eg the Stinnett paper https://pubmed.ncbi.nlm.nih.gov/9566468/).

*** Reviewer #3: 

The authors have addressed my points.

Michael Dewey

***

[LINK]

---

## [Editor Report · Decision Letter 3]

20 Jun 2021

Dear Dr. Menzies,

Thank you very much for re-submitting your manuscript "Economic and modelling evidence for tuberculosis preventive therapy among people living with HIV: a systematic review & meta-analysis" (PMEDICINE-D-20-06011R3) for consideration at PLOS Medicine.

I have discussed the paper with editorial colleagues and our academic editor, and once the remaining editorial and production issues are fully resolved we expect to be able to accept the paper for publication in the journal. The remaining issues that need to be addressed are listed at the end of this email. 

Please let me know if you have any questions, and we look forward to receiving the revised manuscript.   

Sincerely,

Richard Turner, PhD

rturner@plos.org

Requests from Editors:

Please check that numbers are quoted consistently throughout the ms: we note that $271 at line 58 appears to be quoted as $270 at line 441.

Where p values are quoted in table 4, please also quote these numbers alongside the relevant 95% CI in the abstract, e.g., at line 58.

We feel that more cautious language is needed in the abstract, and at the corresponding points in the main text. We note "... found TPT to be more effective" at line 56 and similar wording at line 59. We ask you to convert the phrasing to "... TPT appeared to be more effective" or similar (we did not see statistical tests supporting the arguments of "more effective", and the 95% CI appear to overlap).

At line 84, please add the missing "at" and again adopt more cautious language regarding effectiveness. 

At line 489, again we suggest "apparent effectiveness" or similar.

***

---

## [Editor Report · Decision Letter 4]

27 Jun 2021

Dear Dr Menzies, 

On behalf of my colleagues, I am pleased to inform you that we have agreed to publish your manuscript "Economic and modelling evidence for tuberculosis preventive therapy among people living with HIV: a systematic review & meta-analysis" (PMEDICINE-D-20-06011R4) in PLOS Medicine.

PRESS

Sincerely, 

Richard Turner, PhD 

rturner@plos.org